# Finite Element Analysis and Modeling in Pharmaceutical Tableting

**DOI:** 10.3390/pharmaceutics14030673

**Published:** 2022-03-18

**Authors:** Ioannis Partheniadis, Vasiliki Terzi, Ioannis Nikolakakis

**Affiliations:** 1Department of Pharmaceutical Technology, Faculty of Health Sciences, School of Pharmacy, Aristotle University of Thessaloniki, 54124 Thessaloniki, Greece; ioanpart@pharm.auth.gr; 2Faculty of Engineering, Institute of Static and Dynamic of Structures, School of Civil Engineering, Aristotle University of Thessaloniki, 54124 Thessaloniki, Greece; terziv@civil.auth.gr

**Keywords:** overview, pharmaceuticals, tablet shape, punch shape, simulation, constitutive models, microcrystalline cellulose, Drucker–Prager, compression, density distribution

## Abstract

Finite element analysis (FEA) is a computational method providing numerical solutions and mathematical modeling of complex physical phenomena that evolve during compression tableting of pharmaceutical powders. Since the early 2000s, FEA has been utilized together with various constitutive material models in a quest for a deeper understanding and unraveling of the complex mechanisms that govern powder compression. The objective of the present review paper is to highlight the potential and feasibility of FEA for implementation in pharmaceutical tableting in order to elucidate important aspects of the process, namely: stress and density distributions, temperature evolution, effect of punch shape on tablet formation, effect of friction, and failure of the tablet under stress. The constitutive models and theoretical background governing the above aspects of tablet compression and tablet fracture under diametral loading are also presented. In the last sections, applications of FEA in pharmaceutical tableting are demonstrated by many examples that prove its utilization and point out further potential applications.

## 1. Introduction

Pharmaceutical tablets containing pharmacologically active and functional ingredients are by and large prepared by compression of dry powder mixtures. Tablets are available in different shapes and weights and are mostly administered *per os* to provide immediate or sustained release of the active pharmaceutical ingredient (API) effect. Tablet compression of dry pharmaceutical powders comprises die filling, compaction, and ejection. It is associated with many complex phenomena that can lead to manufacturing problems, such as sticking, picking, capping, and strength variability. Therefore, understanding the behavior of the powder under compression is an important aspect of the tableting process and for defining the properties of the final tablets.

Traditionally, the study of tablet compression is based on two principal equations: the Heckel equation that provides information on the compression stages of the powder [1]; and the hardness equation [2] that gives information on the compactibility of the powder [2,3]. However, these two equations consider only an average stress along the direction of compression, ignoring radial stress transmission and friction. Moreover, they do not address the 3D character of the stress field and the density inhomogeneity inside the tablets, especially when the shape is other than flat-faced [4].

In order to address the limitations of the above equations, constitutive material models representing yield surfaces of compressed powders have been adopted [5,6]. There are several constitutive models available, of which the Drucker–Prager Cap (DPC), the Cam-Clay, and the DiMaggio–Sandler model are better known [7,8,9]. From these, the DPC model is more often employed in pharmaceutical tableting due to its ability to describe the complex physical phenomena taking place during compression, decompression, and ejection [5]. Its basic features and examples of applications in various fields involving powder compaction, including pharmaceuticals, are explicitly described in the review by Sinka (2007) [10]. Modifications of the DPC model addressing specific issues related to the elastoplacticity of pharmaceutical excipients have also been developed, making further contributions in the field [11,12].

Finite element analysis (FEA) by discretization has been implemented as a tool to predict the stress distribution inside the powder under compression. The first attempt to adapt FEA for the study of tablet compression was made by Al-Khattat and Al-Hassani in 1987 [13]. Their numerical simulation model was based on an analogue to the continuum friction model developed by Al-Khattat in 1981 for application in metal’s plasticity [14]. Since the early 2000s, several works have been published utilizing modern FEA models for the study of the compression of pharmaceutical powders, and the number of publications is constantly increasing [12,15,16]. The parameters of the various constitutive models are entered as input in FEA to perform a detailed analysis of tablet compaction and predict tablet properties. Thus, FEA is considered as a practical, potent, resource-saving, and robust mathematical technique for generating crucial information and for predicting the density distribution in the compacted powder, the elastic deformation during tablet ejection, and the tablet strength.

The present review was conducted by using the Scopus^®^ (Elsevier) and the PubMed^®^ (NLM: United States National Library of Medicine) databases. It focuses on the implementation of FEA in pharmaceutical tableting, aiming to enlighten important aspects of this process, namely: tablet failure mechanisms; temperature evolution; stress and density distribution; effect of punch shape and friction on tablet formation; and tensile strength of differently shaped tablets. The constitutive models, their theoretical background governing the different aspects of tableting, and the diametral compression test for the mechanical characterization of tablets (suggested by the United States Pharmacopeia [17]) are exploited, aiming to provide a systematic overview of the latest contributions of FEA and a deeper understanding of the pharmaceutical tableting process.

## 2. Finite Element Modeling and Analysis

### 2.1. Overview

Finite element analysis (FEA) was first introduced in 1943 [18] and has been successfully evolving since then in the field of engineering sciences. The basic ideas behind FEA date back to 1909 [19] and 1915 [20], while the term “Finite Element” was introduced by Clough in 1960 [21].

FEA is a computational tool that involves discretization (meshing) of a domain (body) into a finite number of subdivisions (elements) and mathematical expression of the relevant physical behavior in each element [22]. In the framework of the present review, the primary interest is the strain–stress distributions within the compressed powder during tablet formation and in the final tablet during mechanical testing. Each element is connected to its neighboring elements by points, known as nodes. Nodes connect elements into 2D (plane triangle or quadrangle) or 3D (tetrahedral or hexahedral) shapes [22,23]. Once the appropriate meshing has been established, a constitutive equation describing the physical phenomenon is solved for each node [22,24]. In general, the workflow of FEA consists of the following steps: (i) creation of the geometric model; (ii) meshing; (iii) establishment of boundary conditions; (iv) definition of material properties; (v) definition of loading conditions; (vi) solution of FEA model; (vii) verification and validation of the model [22,24,25,26].

The concepts involved in each step of FEA will be discussed in this section to provide the reader with an overview of the workflow and the individual steps of FEA and enable smooth passage into the main themes related to tablet compression and fracture mechanical test presented in later sections.

### 2.2. Geometry

The first step in FEA is to define the geometry of the system. This is a very important step as it directly affects the quality of the analysis of results. The geometry is usually constructed via computer-aided design (CAD) methods and refers to both the geometry of the compacted powder (flat-faced tablet, biconvex, capsule shape tablet, etc.), the punches, and the die walls of the press. Since during compression the powder bed, the punch, and the die have a vertical axis of symmetry, the boundary conditions are symmetrical with respect to this axis and, hence, it is feasible to create and use only half of the geometry of the compression system. There is no however symmetry on the horizontal axis because during decompression/tablet ejection, the position of the lower punch changes. On the other hand, during mechanical testing by diametral loading, only the quarter of the tablet and platens geometry is modeled since the boundary conditions allow symmetry to both the horizontal and vertical axes. Taking advantage of symmetry is very important since it significantly reduces the computation time without affecting the quality of FEA. Figure 1a depicts 2D symmetry applied for compression, while Figure 1b depicts 2D symmetry applied during diametral compression mechanical strength test. In both cases, a round flat-faced tablet with 8 mm diameter is considered.

### 2.3. Meshing

The next step in FEA is the generation of the mesh, which is basically the process of discretization of the continuous geometry of the system into a network of finite elements connected to one another at nodes. Three different types of finite elements are defined according to the type of equation (also known as shape function) by which they are connected: (i) first order or linear elements, where the nodes are located at the vertices of the elements; (ii) second order or quadratic elements, where they are located at the midpoint of the element edges; (iii) third order elements, where the nodes are located at the midpoint of element faces. Mesh generation is very important. The shape of the constructed element (triangular, quadrilateral, etc.) influences the prediction. The simpler the shape, the better. However, most important for element quality and, consequently, for the resolution and model accuracy and for simulation solving time is the element size. Improper size can lead to erroneous calculations and inaccurate solutions of the FEA. Mesh convergence or mesh sensitivity studies are usually performed in preliminary stages of analysis to determine the optimal number and size of the comprising elements. In tablet compression and strength test analyses, the 4-node quadrilateral bilinear element is usually applied.

### 2.4. Boundary Conditions

Boundary conditions are constraints or restraints imposed on the degrees of freedom of the parameters of the mechanical system and also on the friction or interface conditions between the punches and die walls. The main purpose of the constraints is to restrict motion (translation and/or rotation) at each node, i.e., to limit possible, permissible movements. The main boundary conditions that are imposed during compression simulation are (i) the upper punch is constrained to move vertically along the y-axis with specified compression speed; (ii) the lower punch is either constrained to move vertically along the y-axis or its translational and rotational degrees of freedom are fixed; in the case that tablet ejection is not taken into account or if it follows a different mechanism, the above conditions are imposed separately; (iii) the degree of freedom of the die walls is fixed; (iv) a constant friction coefficient (*μ*) boundary condition is applied at the powder/tooling interface (*μ* values usually range between 0.1 and 0.35 [5,27]). Regarding the test of mechanical strength, the main boundary condition is the vertical movement of the platen along the y-axis.

### 2.5. Material Properties

The fourth step defines the properties of the material comprising the tablet and their assignment to finite elements. The definition of material properties depends on the type of analysis and the conceptualization of the problem. In linear analysis, under the assumption of an isotropic elastic material, the Young’s modulus (*E*) and Poisson’s ration (ν) have to defined. For nonlinear analyses, the yield stress is added. When thermal effects are investigated (e.g., temperature evolution during tableting), the thermal conductivity (k) and specific heat (*C*_p_) are defined additionally. The tooling bodies (punches and die walls) are usually modeled as rigid bodies. Otherwise, the properties of steel are adopted (*E* = 220 GPa and *ν* = 0.3 [28]).

### 2.6. Loading Conditions

Ιn FEA, loading is entered through the nodes of the model mesh. There are many types of loading conditions available that can be used to study the performance of the model: (i) concentrated forces/moments; (ii) distributed forces/moments; (iii) nodal displacements/velocities/accelerations; (iv) thermal loads. Loading is very important as it is applied as input data to the chosen constitutive equation and defines the model response during the stage of model solution. For ultimate tablet fracture parametric analysis, it is usually applied using a range of static loads for the determination of the force required to generate sufficient fracture stresses. On the other hand, for the simulation of compression, dynamic loading conditions are used. In this case, the loading rates are also important since pharmaceutical powders are viscoelastic and their compressive behavior is sensitive to strain rates induced by the compression speed [29].

In some cases, and especially during the mechanical strength test simulation, it is tempting to model the applied external forces at a single point of the geometry. However, these point loads can be problematic and should be avoided because they are interpreted by the FEA as application of a finite force over an infinite small area, leading to infinite stresses applied at this point. By increasing mesh density, the stresses will also increase, resulting in singularities in the model.

### 2.7. FEA Model Solution

Once the prerequisite steps have been fulfilled, the FEA model is solved by implementing constitutive equations. These characterize the material and its behavior under the applied loads and describe the macroscopic performance resulting from the internal material characteristics [30].

Firstly, the applied constitutive equation has to be calibrated. This means that model parameters have to be estimated first experimentally (e.g., by using instrumented tableting equipment). Then, the known parameters of the equation for the particular material in question are used together with the material properties and the constraints for every node of the mesh to solve the FEA model. All of these are combined in a global stiffness matrix, [*K*], which relates the displacement with the applied loads, or the resistance to displacement. The applied loads are combined in a vector, {F}, (Equation (1)), to calculate the primary variable, which is the nodal displacements vector, {u}.
(1)[K]·{u}={F}

Equation (1) refers to static analyses. For dynamic analyses, the equation of motion is used, which has the following form:(2)$$\left[M\right]\cdot \left\{\ddot{u}\right\}+\left[C\right]\cdot \left\{\dot{u}\right\}+\left[K\right]\cdot \left\{u\right\}=\left\{F\right\}$$
[M], [C] are the mass and damping matrices, respectively, and $\left\{\ddot{u}\right\},\;\left\{\dot{u}\right\}$ are the acceleration and displacement vectors. The mass matrix refers to the inertial characteristics of the studied specimens, while the damping matrix encompasses the intrinsic damping mechanisms of the system.

The calculated primary variable vector, {u}, is used to determine the derived variables of interest, that is, the stress and the strain. In linear analyses, [K] matrix is constant and the displacement is proportionally correlated to the applied loads. In nonlinear analyses, the material behavior due to the applied load is non-linear dependent on the inflicted displacement and the compacted powder is modeled as elastoplastic entity. In this case, [K] and {F} are functions of {u}, which necessitates application of an iterative approach [25].

### 2.8. FEA Model Verification and Validation

Verification and validation (V&V) of the FEA model are very important as they ensure its accuracy in terms of model assumptions, meshing quality, boundary, and loading conditions. There are several guidelines for the V&V of FEA models. For solid mechanics, the most important are those issued by the American Society of Mechanical Engineers (ASME) [31]. In Figure 2, a V&V flowchart based on the ASME 10.1 Standard is presented. The first step is verification of the code. This process identifies and removes any bugs or programming errors generated while implementing the numerical algorithms. Modern FEA can be easily accessed through several software packages, e.g., Abaqus (Dassault Systems, Johnston, RI, USA), Ansys (ANSYS Inc., Canonsburg, PA, USA), and COMSOL (Burlington, MA, USA). These software packages provide benchmark manuals explaining the verification of the implemented algorithms since “code verification” is their responsibility. The next step is “calculation verification”, which is carried out by the user/researcher. It involves the quantification of errors due to insufficient mesh discretization, improper boundary conditions, approximations in material properties, and errors during model generation. The most common source of errors during model generation is mesh discretization and, for this reason, mesh convergence studies are commonly performed in preliminary FEA stages. Usually, the FEA model validation initiates with the creation of a simple model that can be confirmed via basic analytical solutions. Further validation involves comparison of the available experimental and numerical results in order to test the ability of the simulation to capture the physical behavior of the experimental process. As soon as the results converge, more complex models can be designed and tested, or even parametric analyses can be conducted.

## 3. Theoretical Background: Constitutive Equations

In this section, the constitutive equations that are used to describe tablet compression or tablet mechanical strength will be presented. The parameters derived from these equations are subsequently implemented as inputs in the FEA model to perform a detailed analysis and gain information on the internal structure of the compacted powder or final tablet.

### 3.1. Tableting

#### 3.1.1. Powder as Continuum Medium

During powder compression, several interacting processes take place, and the extent of the involvement of each is difficult to quantify and predict [2,3]. In the first stage of compression, powder is fed into the die to form a loosely packed bed. At low compression, particle rearrangement takes place. At high compression, the powder densifies by friction, fragmentation, deformation, and mechanical particle interlocking. Therefore, in terms of the continuum model, powder compression can be regarded as an irreversible deformation from the state of low porosity (loose powder) to high porosity (compact).

Since the compressed particles are several orders of magnitude smaller than the physical size of the tablet, the powder can be regarded macroscopically as a continuous, porous medium and a representative volume medium consisting of numerous particles can be defined. This medium represents macroscopic responses and should be insensitive to variations at the particle level [4]. Therefore, this approach does not consider the characteristics of individual particles, but the powder is treated as a continuum medium characterized by averaging parameters (e.g., cohesion, interparticle friction).

In terms of continuum medium, the evolution of microstructure and loading/deformation can be studied using FEA together with constitutive mechanical models. These models describe the deformation under stress and the friction between the material and the tools of the press (punch dies) [4,5] and include density-dependent parameters. For the analysis, the combined averaging parameters derived from the macroscopic responses are entered into the density-dependent parameters of the constitutive models. Additionally, knowledge of the geometry of the die and punches and of the sequence of punch motion is necessary for the analysis [10].

Several constitutive models based on the elastoplastic theory have been adopted for the compression of porous materials [7,8,32,33,34]. The most common are presented next, in order of increasing complexity. Emphasis is placed on the Drucker–Prager Cap (DPC) model and its modifications.

#### 3.1.2. Linear Elasticity Model

This model is essential for understanding the behavior of a material during tablet ejection since, at this stage, the strains are predominantly elastic. In general, the elastic behavior of powders is neither isotropic nor perfectly linear. However, to simplify the study of material behavior during ejection, isotropic material with linear elastic behavior is assumed. The elastic strains and stresses are described by the following equations:(3)$$\varepsilon_{ii}=\frac{\sigma_{ii}}{E}-\frac{v}{E}\cdot \overset{}{\underset{j\neq 1}{\sum }}\sigma_{jj}$$
(4)$$\gamma_{ij}=\left(1+v\right)\cdot \frac{\sigma_{ij}}{G}$$
(5)$$G=\frac{E}{2\cdot \left(1+v\right)}$$
where $\varepsilon_{ii}$ and $\gamma_{ij}$ are the normal and the shear strains, respectively, E is the Young’s modulus, v the Poisson’s ratio, and G the shear modulus. Therefore, the behavior of a linear isotropic elastic material is fully explained by the parameters E and v [35,36].

#### 3.1.3. Linear Viscoelasticity Model

This model is implemented when the quality of the final tablet depends on the loading rate (compression speed), in other words, when there is strain rate sensitivity (SRS) [37,38,39]. In this case, the applied stress (σ) is time-dependent and is expressed by Equation (6) [40]:(6)$$\sigma \left(t\right)=\int_{0}^{t}E\left(t-\tau \right)\cdot \frac{d\varepsilon \left(\tau \right)}{d\tau }d\tau$$
where *t* is the time and *τ* a time variable of integration. The stress–strain relationship is a function of the loading history (*t* = 0 corresponds to the beginning of loading). Equation (6) represents the case of uniaxial loading. For three-dimensional loading, the above approach is extended using the volumetric and deviatoric parts of stress and strain [41], which suggests the use of bulk (K) and shear (G) moduli. The deviatoric ($q^{\prime }$) and hydrostatic (P) stresses are expressed by Equations (7) and (8):(7)$$q^{\prime }\left(t\right)=\int_{0}^{t}G\left(t-\tau \right)\cdot \frac{d\varepsilon_{s}\left(\tau \right)}{d\tau }d\tau$$
(8)$$P\left(t\right)=\int_{0}^{t}K\left(t-\tau \right)\cdot \frac{d\varepsilon_{v}\left(\tau \right)}{d\tau }d\tau$$
where $\varepsilon_{s}$ and $\varepsilon_{v}$ are the deviatoric and volumetric strains, respectively. Application of Equations (7) and (8) prerequisite an analytical expression of the time dependency of (K) and (G). For this reason, the Prony series, derived from the generalized Maxwell stress relaxation model, is used [42,43]. The G(t) and K(t) are expressed by Equations (9) and (10):(9)$$G\left(t\right)=G_{\infty }\cdot \left(1+\overset{n}{\underset{i=1}{\sum }}G_{i}\cdot e^{-t/\tau_{i}}\right)$$
(10)$$K\left(t\right)=K_{\infty }\cdot \left(1+\overset{n}{\underset{i=1}{\sum }}K_{i}\cdot e^{-t/\tau_{i}}\right)$$
where $G_{\infty }$ and $K_{\infty }$ are the infinite/long-term shear and bulk moduli; and $G_{i}$, $K_{i}$, and $\tau_{i}$ are model parameters.

#### 3.1.4. Elastoplastic Models

There are two major phenomenological theories of plasticity: (i) the incremental and (ii) the deformational [44]. The incremental theory considers dependency of material deformation on the history of loading, while the deformational considers independency of the deformation on loading. Since the incremental theory of plasticity is more representative to real cases, it has been implemented in the description of plastic deformation of a wide range of materials [4,10,44,45,46].

In general, three parameters of a constitutive model have to be defined for materials that experience microstructural changes during deformation. The first is the yield parameter, which describes the transition from elastic to plastic behavior, the second is the plastic potential, which connects the yield condition and the flow rule of the plastic components, and the third is the evolution of microstructure that defines the resistance to further deformation. The ways that these parameters contribute to the constitutive model are discussed in accordance with the elastoplastic model.

In the elastoplastic model, the total increment strain tensor, $d\varepsilon_{ij}$, results from the contribution of the elastic strain increment (reversible), $d\varepsilon_{ij}^{el}$, and the plastic strain increment (irreversible), $d\varepsilon_{ij}^{pl}$, as described by Equation (11):(11)$$d\varepsilon_{ij}=d\varepsilon_{ij}^{el}+d\varepsilon_{ij}^{pl}$$
where subscripts i and j vary between 1 and 3 depending on the coordinate system axes, indicating the 3D nature of strain. For simplification, the contribution of increment is assumed to be additive [4,15]. A multiplicative contribution of each increment has also been developed and could give better accuracy, but it is quite complex and is not described here [47,48,49].

The $d\varepsilon_{ij}^{el}$ component of Equation (11) represents the reversible part of the deformation and describes the elastic recovery of the material during the decompression stage of tableting. The relationship between the macroscopic stress and the elastic strain is described by a linear Hooke’s Law (Equation (12)) [36]:(12)$$d\sigma_{ij}=\overset{3}{\underset{k,l=1}{\sum }}L_{ijkl}\cdot d\varepsilon_{kl}^{el}$$
where $d\sigma_{ij}$ is the three-dimensional form of the stress and $L_{ijkl}$ is the elastic stiffness. For isotropic materials, the (3 × 3 × 3 × 3) components of $L_{ijkl\;}$ contain only two independent parameters (Young’s modulus and Poisson’s ratio) [4].

The $d\varepsilon_{ij}^{pl}$ component of Equation (11) represents the irreversible part of deformation and is related to particle rearrangement, fragmentation, and plastic deformation [4].

The first parameter that needs to be defined is the yield parameter, which sets the limits between elastic and plastic deformation, as expressed by the yield function $F\left(\sigma_{ij},\;k_{m},\;\cdots \right)$ in Equation (13):(13)$$F\left(\sigma_{ij},\;k_{m},\;\cdots \right)=\{\begin{array}{c} <0,\;\mathrm{elastic}\;\mathrm{deformation}\\ =0,\;\mathrm{plastic}\;\mathrm{deformation} \end{array}\;$$
where $\sigma_{ij}$ is the stress and $k_{m}\;\left(m=1,\cdots ,n\right)$ internal state variables affecting the yield parameter of the material under compaction. The second parameter that needs to be defined is the plastic flow potential $d\sigma_{ij}^{pl}$, which determines the three-dimensional form of plastic deformation as expressed by Equation (14):(14)$$d\sigma_{ij}^{pl}=d\lambda \cdot \frac{\partial G\left(\sigma_{ij},\;k_{m},\;\cdots \right)}{\partial \sigma_{ij}}$$
where $G\left(\sigma_{ij},\;k_{m},\;\cdots \right)$ is the flow function and dλ is a parameter for hardening.

Equations (11), (12), and (14) are given in an incremental fashion since the state of the material evolves progressively during compression and, therefore, can describe the complete continuum mechanics constitutive model.

The evolution of the microstructure, which is the third parameter of interest, is the relative density (or solid fraction, $p_{F}$) of the compacted material, which is the ratio of the density of the powder under compaction at a given state over the material true density.

The yield parameter and the plastic flow potential depend on two stress invariants: (i) the hydrostatic pressure, P, and (ii) the von Mises equivalent or effective stress, q [4,50]. These are associated with the volumetric plastic strain and distortion, respectively, and are expressed by Equations (15) and (16):(15)$$P=-\frac{1}{3}\cdot \left(\sigma_{11}+\sigma_{22}+\sigma_{33}\right)$$
(16)$$q={\left\{\frac{3}{2}\cdot \left[{\left(\sigma_{11}+P\right)}^{2}+{\left(\sigma_{22}+P\right)}^{2}+{\left(\sigma_{33}+P\right)}^{2}+2\cdot \sigma_{12}^{2}+2\cdot \sigma_{23}^{2}+2\cdot \sigma_{13}^{2}\right]\right\}}^{1/2}$$

#### 3.1.5. The Drucker–Prager Cap (DPC) Model

Drucker–Prager Cap (DPC) is the constitutive model that is often used to study tablet compaction. It is able to describe the complex events taking place during compression, namely: volume reduction, shear flow, nonlinear densification due to frictional forces, stress hardening, and elastic recovery occurring during decompression and ejection [5]. The model is pressure-dependent and assumes isotropic powder behavior [5,8].

Figure 3a depicts a 2D projection of DPC model at a fixed solid fraction. The Y-axis corresponds to the von Mises equivalent or effective stress, q, and the X-axis corresponds to the hydrostatic pressure, P. The two yield surfaces represent two types of deformation under compression. At high hydrostatic pressures (advanced compaction), the material densifies when the pressure exceeds a certain value, *P*_a_, as represented in Figure 3a (right) by the “cap” line ($F_{c}$). At low hydrostatic pressures, the material exhibits shear failure, as represented in Figure 3a (left) by the failure line ($F_{s}$). Therefore, at the beginning of compaction, the DPC is a shear failure model reflecting the dependence of strength on the intrinsic cohesion and angle of internal friction of the powder [4,5,8]. At low pressures, the DPC model predicts that the strength in tension (negative *P* values) is smaller than in compression, which is common for rocks [4,51]. The straight line ($F_{s}$), also known as Mohr Coulomb shear failure line, is defined by Equation (17) [8,51,52,53,54]:(17)$$F_{s}\left(q,P\right)=q-d-P\cdot tan\left(\beta \right)$$
where d is the cohesion and β the angle of internal friction. According to Equation (17), when the combination of hydrostatic pressure stress, P, and the von Mises equivalent stress, q, results in $F_{s}\left(q,P\right)<$0, only elastic deformation occurs. On the other hand, when P and q result in $F_{s}\left(q,P\right)=$ 0, the material fails in shearing.

At the advanced compaction phase, the yield surface is described by the “cap” line on the right of Figure 3a. The elliptical line, $F_{c}$, describes the plastic behavior strain-hardening of the material under high hydrostatic pressure [5,8] and is defined by Equation (18) [4,5,8]:(18)$$F_{c}\left(q,\;P\right)=\sqrt{{\left(P-P_{a}\right)}^{2}+{\left(\frac{R\cdot q}{1+a-\frac{a}{cos\left(\beta \right)}}\right)}^{2}-R\cdot \left(d+P_{a}\cdot tan\left(\beta \right)\right)}$$
where $P_{a}$ is an evolution parameter representing the volumetric inelastic strain driven hardening/softening, R is an eccentricity parameter that defines the shape of the “cap”, and a is a transition surface radius parameter with values in the range 0.01 to 0.05 [4,5]. $\;P_{a}$ and R are estimated from compaction experiments during the model calibration procedure. Parameter a does not have physical meaning. It is used to avoid the formation of a corner, which may lead to numerical problems at the transition surface segment ($F_{t}$), where $F_{s}$ and $F_{c}$ intersect. In Figure 3, parameter a has a zero value and, therefore, the ($F_{t}$) segment is depicted as a corner.

Full description of the DPC model requires definition of the flow potential. Two types are defined by the DPC model: (i) the associated flow potential, $G_{c}$, corresponding to the “cap” region, and (ii) the plastic flow potential, $G_{s}$, corresponding to shear line and the transition segment ($F_{t}$). In the “cap” region, the associated flow potential function, $G_{c}$, and the yield surface function, $F_{c}$, coincide. This behavior is termed ‘associated plasticity’ [55,56] and is expressed by equality (Equation (19)):(19)$$G_{c}=F_{c}=0$$

On the other hand, the plastic flow potential function, $G_{s}$, is non-associated ($G_{s}\neq F_{s}$) and is expressed by Equation (20):(20)$$G_{s}=\sqrt{{\left[\left(P-P_{a}\right)\cdot tan\left(\beta \right)\right]}^{2}+{\left(\frac{q}{1+a-\frac{a}{cos\left(\beta \right)}}\right)}^{2}}$$

$G_{c}$ and $G_{s}$ predict the densification in the corresponding surfaces when the hydrostatic pressure is above and below $P_{a}$, respectively (Figure 3a).

The yield surface function, $F_{s}\left(q,P\right)$, and the flow potential function receive unique values for a given level of relative density, $p_{F}$. Complete description of material behavior during compression at different $p_{F}$ values requires collection of a family of DPC models (Figure 3b). As the powder bed gets denser during compression, the applied stresses on the material are represented by the yield surface corresponding to the current, $p_{F}$. As $p_{F}$→ 1, the DPC model is consistent with the fully dense material behavior, that is, a Mohr–Coulomb line at low hydrostatic pressures and no pressure dependence of the yield function at high hydrostatic pressures [12].

The DPC also uses a hardening law to describe the strain hardening behavior during compression. According to this law, the hydrostatic compression yield stress, $P_{b}$, depends on the volumetric plastic strain, $\varepsilon_{v}^{pl}$, according to Equation (21). The volumetric plastic strain is expressed by Equation (22).
(21)$$P_{b}=f\left(\varepsilon_{v}^{pl}\right)$$
(22)$$\varepsilon_{v}^{pl}=In\left(p_{F}^{\prime }/p_{F0}\right)$$
where $p_{F}^{\prime }$ and $p_{F0}$ are the final and initial relative densities, respectively.

Using microcrystalline cellulose (MCC) as a model material, Sinka et al. (2001) [16] adopted a modified Drucker–Prager cap model where the elastic and plastic model parameters are expressed as functions of relative density. The parameters of the model were determined using an instrumented die. Young’s modulus and Poisson’s ratio were determined from the unloading curve of the material compressed to a given density.

#### 3.1.6. Constitutive Model for Plastic Materials

Edmans and Sinka (2020) [11] introduced the von Mises Double Cap (VMDC) model to describe the behavior of plastically compressible spherical particles. The model discerns the compressive, deviatoric, and dilatory plastic behavior and is particularly useful for the analysis of dense particulate systems where particles deform greatly under stress. The parameters used for the VMDC model were based on experimental data obtained for common pharmaceutical excipients, and the model parameters can be predicted from material. The model can be employed in discrete element to predict macroscopic properties of porous granular materials, while the analytical framework and qualitative findings can be used in the design of granules.

The load-displacement response during the loading stage is represented by the 3-parameter Equation (23) in which an initially linear response is combined with an exponential hardening response.
(23)$$\;\;\;\;\;\;\;\;\;\;\;\;\;\;\;\;\;\;\;\;\;\;\;\;\;\;\;\;\;\;\;\;\;\;\;F_{model}\left(\delta \right)=\{\begin{array}{c} \kappa \delta ,\;\;\delta \;<\;\delta t\;\\ \kappa \delta texp\left(\alpha \left(\delta -\delta t\right)\right),\delta <\delta <\delta max\;\; \end{array}$$

κ is dimensionless initial stiffness, *δ* dimensionless normal particle displacement. *δmax* has the value 0.5 limited by the simulation data.

To describe the non-linear load-displacement response during the unloading processes, separate parameters representing the nonlinearity of the unloading curve and the relative unloading stiffness were defined as in Equation (24):(24)$$Funl\left(\delta \right)=\{\begin{array}{l} 0,\;\delta \;<\;\delta o\;\\ \kappa {\left(\delta -\delta ο\right)}^{\alpha },\;\delta \;>\;\delta ο\; \end{array}$$
where F is the load, *δ*_o_ the nondimensional displacement at separation, the nonlinearity exponent, α, is 1.5, and *k* is given and κ is a function of Young’s modulus and Poisson’s ratio.

#### 3.1.7. Powder–Wall Friction Law

During compression, friction between the powder and die wall induces non-uniform axial stresses that lead to density gradients [57,58] and, hence, ejection forces and defects in the final tablets [12]. Therefore, incorporation of the powder–wall friction law in the FEA model and determination of the friction coefficient, μ, is essential for the study of failure mechanisms. According to the Janssen–Walker model, the friction coefficient, μ, is defined by Equation (25) [57,59]:(25)$$\mu =\frac{D}{4\cdot h}\cdot \frac{\sigma_{Β}}{\sigma_{r}}\cdot {\left(\frac{\sigma_{Τ}}{\sigma_{Β}}\right)}^{z/h}\cdot In\left(\frac{\sigma_{Τ}}{\sigma_{Β}}\right)$$
where D is the internal diameter of the die, h the height of the compact, $\sigma_{r}$ the radial compression stress at position z from the compact surface, and $\sigma_{Β}$ and $\sigma_{Τ}$ are the axial compression stresses applied by the upper and lower punch, respectively. Although this model can accurately calculate the friction coefficient, it requires elaborate experimental set-up, and, for this reason, it is often bypassed and a value of μ = 0.1 is assigned [5,60,61].

Sinka et al. (2001) [16] applied a modified Drucker–Prager cap model for the study of the effect of wall friction in the compaction of MCC tablets with curved surfaces using instrumented die fitted with radial sensors. They found that the friction coefficient depended on contact pressure and, starting from high values at the early stages of compaction, it asymptotes to a lower value as the contact pressure and relative density are increased. They also demonstrated that tablets of the same material, identical in shape, may have different microstructures depending on the existing frictional conditions. This is because high density regions develop around the edge of the tablet due to the curvature of the punch.

### 3.2. Mechanical Strength Test

From the available methods of the determination of the tablet mechanical strength, the diametral compression test is most commonly employed [62]. It was independently developed by Barcellos and Carneiro [63] in Brazil (known as “Brazilian” test) and by Akazawa [64] in Japan for testing concrete specimens. It is also known as “indirect” tensile test because fracture results from compressive loads [65].

Consideration of the elastic stresses in a flat-faced tablet subjected to two concentrated diametral loads during mechanical strength testing is essential for modeling and studying the fracture mechanism. According to Timoshenko’s theory of elasticity [36], the stress field is expressed by Equations (26)–(28) [62,66,67,68].
(26)$$\sigma_{xx}\left(x,y\right)=-\frac{2\cdot P}{\pi \cdot t}\cdot \left\{\frac{\left(\frac{\Phi }{2}-y\right)\cdot x^{2}}{{\left[{\left(\frac{\Phi }{2}-y\right)}^{2}+x^{2}\right]}^{2}}+\frac{\left(\frac{\Phi }{2}+y\right)\cdot x^{2}}{{\left[{\left(\frac{\Phi }{2}+y\right)}^{2}+x^{2}\right]}^{2}}-\frac{1}{\Phi }\right\}$$
(27)$$\sigma_{yy}\left(x,y\right)=-\frac{2\cdot P}{\pi \cdot t}\cdot \left\{\frac{{\left(\frac{\Phi }{2}-y\right)}^{3}}{{\left[{\left(\frac{\Phi }{2}-y\right)}^{2}+x^{2}\right]}^{2}}+\frac{{\left(\frac{\Phi }{2}+y\right)}^{3}}{{\left[{\left(\frac{\Phi }{2}+y\right)}^{2}+x^{2}\right]}^{2}}-\frac{1}{\Phi }\right\}$$
(28)$$\tau_{xy}\left(x,y\right)=\frac{2\cdot P}{\pi \cdot t}\cdot \left\{\frac{{\left(\frac{\Phi }{2}-y\right)}^{2}\cdot x}{{\left[{\left(\frac{\Phi }{2}-y\right)}^{2}+x^{2}\right]}^{2}}+\frac{{\left(\frac{\Phi }{2}+y\right)}^{2}\cdot x}{{\left[{\left(\frac{\Phi }{2}+y\right)}^{2}+x^{2}\right]}^{2}}\right\}$$
where $\sigma_{yy}$ is the compressive stress; $\sigma_{xx}$ is the tensile stress; $\tau_{xy}$ is the shear stress; P is the concentrated (acting on a relatively small area) compressive load; Φ is the diameter of the tested tablet; t is its thickness; x, y are the coordinates of a point that belongs to the surface of the tablet according to a rectangular coordinate *Oxy* system (where *O* coincides with the center of the tablet).

At the center of the tablet (x=0, y=0) Equations (26) and (27) can be reduced to Equations (29) and (30), respectively. (Equation (28)) and the shear stress $(\tau_{xy}$) equals zero.
(29)$$\sigma_{xx}\left(0,0\right)=\frac{2\cdot P}{\pi \cdot \Phi \cdot t}$$
(30)$$\sigma_{yy}\left(0,0\right)=-3\cdot \frac{2\cdot P}{\pi \cdot \Phi \cdot t}=-3\cdot \sigma_{xx}\left(0,0\right)$$

According to the theory, upon breakage, the vertical compressive stress in the center of the tablet is three times larger than the horizontal tensile stress. Modifications of the above equations are often used in the literature for the calculation of the tensile strength of various tablet shapes, e.g., flat-faced [69], doubly convex cylindrical shaped [70,71], capsule shaped [72,73]. However, to establish an FEA model for mechanical strength test, the equations describing the whole stress field in the tablet (Equations (26)–(28)) are necessary. Figure 4 depicts the field of stresses developing at the end of a diametral test.

### 3.3. Limitations of the Constitutive Models in the Pharmaceutical Field

The constitutive models described here are generated for the study of the compression behavior of metallic powders. However, pharmaceutical powders are characterized by significantly lower particle densities compared to metallic powders, having a true density in the region of 1.0 to 1.7 g/cc. In addition, the relative density of pharmaceutical powders at the beginning of the compression process (0.3–0.4) is also lower compared to that of metallic or ceramic powders [10]. Moreover, the compression of powders is a very fast procedure, with a duration of only a few milliseconds [10,38]. Therefore, air and powder interaction become of importance and, as the air pressure raises within the pores of the powder bed, it can reach to values that are comparable with the local strength of the material [10]. Thus, a better understanding of the processes is of importance, requiring insight into the complex phenomena occurring at various length scales from particle contacts to the final pharmaceutical tablet, and, therefore, improved constitutive models, providing better understanding of the compression behavior, have been developed in the last years that are expected to provide a multi-scale modelling approach [11].

## 4. Application of FEA Modelling for Pharmaceutical Tablets

In this section, examples of the application of FEA modeling in the tableting and mechanical test will be presented. The experimental materials, equipment used, the properties of the FEA models, the objectives of the studies, and the literature sources are summarized in Table 1.

The largest number of the presented studies (35.6% of the total) deal with the stress/density distribution within the tablet during compression [5,15,27,61,74,75,76,77,78,79,80,81,82,83,84,85]. From these, 31.3% deal with the effect of punch shape [61,74,79,82,85]. The second most studied aspect (22.2% of the total) is the stress/strain distribution in the tablet during mechanical strength testing [65,81,86,87,88,89,90,91,92,93]. From these, 40% deal with the effect of tablet shape and studied crack propagation [89,90,92,93]. Two studies (4.4% of the total) deal with the stress/strain distribution during the 3-point bending test [94,95]. The rest of the studies deal specifically with failure mechanisms (15.6%) [96,97,98,99,100,101,102], friction (11.1%) [4,28,103,104,105], temperature evolution (6.7%) [106,107,108], and viscoelastic behavior (6.7%) [29,109,110]. From the above account, it appears that FEA has been mostly implemented for the study of stress distribution during tablet formation by compression and during testing of various tablet shapes. More complex aspects, e.g., temperature evolution and viscoelastic behavior, have received less attention. Therefore, FEA is not yet adequately explored in the area of pharmaceutical tableting despite its use since 2002.

Moreover, Table 1 indicates the applicability of Drucker–Prager Cap (DPC) model and its ability to describe complex physical phenomena taking place during tableting since it is utilized in the majority (73.3%) of the studies. It is also noticeable that, in most of the studies (44.4%), microcrystalline cellulose (MCC) is used. This is attributed to its excellent compressibility, the abundant literature data available, and its wide use in the pharmaceutical industry.

**Table 1 pharmaceutics-14-00673-t001:** Studies on the application of finite element analysis (FEA) in the tableting and mechanical strength test of pharmaceutical tablets.

Study	Material ^1^	Equipment	FEA Model			Application	References
			Model ^2^	Meshing ^3^	Young’s Modulus (*E*, GPa)/Poisson Ratio (*ν*, -)		
01	LAC	flat-faced punches	DPC	4-node	4.6/0.17 *	Stress & density distribution during tableting	Michrafy et al., 2002 [15]
02	MCC	9.525 mm die, flat-faced punches	DPC	no data	*E* & *ν* expressed as function of relative density	Stress & density distribution/friction evolution during tableting	Cunningham et al., 2004 [4]
03	MCC	flat-faced punches	DPC & Janssen–Walker model	4-node	*E* expressed as function of relative density/0.18	Study of friction evolution during tableting	Michrafy et al., 2004 [104]
04	MCC	25 mm die, concave-faced punches	DPC & Janssen–Walker model	4-node	*E* & *ν* expressed as function of relative density	Study of friction evolution during tableting	Sinka et al., 2004 [105]
05	MCC	11.28 mm die, flat-faced punches	DPC	4-node	*E* expressed as function of relative density/0.18	Stress & density distribution during tableting	Kadiri et al., 2005 [80]
06	LAC	8 mm die, flat-faced punches	DPC	4-node	3.57/0.12 *	Study of failure mechanisms	Wu et al., 2005 [102]
07	MCC	diametral compression of flat-faced tablets	Filon theory	4-node	1.0/0.25	Stress & strain distributions byopposing compressive line loads	Drake et al., 2007 [87]
08	MCC	8 mm die, flat-faced punches	DPC	no data	no data	Stress & density distribution/failure mechanisms during tableting	Han et al., 2008 [75]
09	MCC	8 mm die, flat- & concave-faced punches	DPC	4-node	4.2 and 22.6/0.42 and 0.233 *	Stress & density distribution during tableting	Han et al., 2008 [76]
10	LAC	8 mm die, flat- & concave-faced punches	DPC	4-node	3.57/0.12	Study of failure (capping) mechanisms	Wu et al., 2008 [101]
11	MCC	9.525 mm die, flat-faced punches	DPC	4-node (T)	no data	Study of temperature evolution during tableting	Zavaliangos et al., 2008 [108]
12	MCC	9.525 mm die, concave-faced punches	DPC	8-node (T)	*E* & *ν* expressed as function of relative density	Study of temperature evolution during tableting	Klinzing et al., 2010 [106]
13	LAC	flat-faced punches	DPC	4-node (R)	no data	Stress & density distribution during tableting	Sinha et al., 2010 [84]
14	MCC	flat-faced punches	DPC	4-node	2.207/0.14 *	Stress & density distribution during tableting	Sinha et al., 2010 [27]
15	LAC	5.6 mm die, flat-faced punches	DPC	4-node	4.86/0.12 *	Stress & density distribution during tableting	Si and Lan, 2012 [83]
16	MCC	flat-faced punches	DPC & creep behavior model	no data	no data	Study of the viscoelastic behavior during tableting	Diarra et al., 2013 [109]
17	LAC, CS & MCC	8 mm die, flat-faced punches	DPC	no data	3–4/0.1–0.2	Stress & density distribution during tableting	Hayashi et al., 2013 [77]
18	MCC	11.28 mm die, flat & concave-faced punches	DPC	4-node (R)	*E* & *ν* expressed as function of relative density	Study of failure (capping) mechanisms	Kadiri and Michrafy, 2013 [99]
19	not applicable/theoretical study	diametral compression of elongated tablets	Elastic stresses model	3-node	0.002/no data	Stress & strain distributions during diametral compression	Pitt and Heasley, 2013 [65]
20	Anhydrous dextrose	diametral compression of flat-faced tablets	Elastic stresses model	20-node (R)	2.58 and 9/0.35 and 0.3	Stress & strain distributions during diametral compression	Podczeck et al., 2013 [89]
21	MCC	diametral compression of flat-faced and biconvex tablets	DPC & Elastic model	8-node	*E* & *ν* expressed as function of relative density	Stress & strain distributions during diametral compression	Shang et al., 2013 [92]
22	MCC	9.525 mm die, flat-faced punches	Griffith & Irwin models	4-node	no data	Study of failure (cracking) mechanisms during decompression & ejection	Garner et al., 2014 [98]
23	TEO, LAC, CS, MCC, MgSt	8 mm die, flat-faced punches	DPC	no data	6.51–9.84/0.1164–0.1282	Stress & density distribution during tableting	Hayashi et al., 2014 [78]
24	MCC	5.25 mm die, flat- & convex-faced punches	DPC	8-node	*E* & *ν* expressed as function of relative density	Stress & density distribution during tableting	Krok et al., 2014 [61]
25	ACP	three point bending test of flat-faced tablets	Elastic stresses model	4-node	3.4/0.23	Stress & strain distribution during three point bending test	Mazel et al., 2014 [94]
26	not applicable/theoretical study	diametral compression of flat, round, bevel-edged tablets	Elastic stresses model	4-node	2.58/0.35	Stress & strain distributions during diametral compression	Podczeck et al., 2014 [90]
27	ACP & MCC	11.28 mm die, flat- & concave-faced punches	DPC	no data	no data	Stress & density distribution during tableting	Diarra et al., 2015 [74]
28	MCC	diametral compression of biconvex tablets	DPC	no data	*E* & *ν* expressed as function of relative density	Study of failure (capping) mechanisms	Furukawa et al., 2015 [97]
29	ACP	11.28 mm die, concave-faced punches	DPC	no data	no data	Stress & density distribution/friction evolution during tableting	Mazel et al., 2015 [28]
30	ACT, LAC, CS, MCC, L-HPC, MgSt	12 mm die, concave-faced punches	DPC	no data	2.96–6.51/0.0742–0.0943	Stress & density distribution during tableting	Otoguro et al., 2015 [82]
31	LAC, ASA	three point bending test of flat, round, bevel-edged tablets	Elastic & brittle-cracking model	10-node	2.99 for LAC & 1.51 for ASS/0.3	Stress & strain distributions during three point bending test	Podczeck et al., 2015 [95]
32	MCC	8 mm die, flat- & convex-faced punches	DPC	4-node (T)	*E* & *ν* expressed as function of relative density	Study of temperature evolution during tableting	Krok et al., 2016 [107]
33	ACP, CPD, SD-LAC, G-LAC & SD-MAN	3.8 mm die, flat-faced punches	DPC	no data	*E* & *ν* expressed as function of relative density	Stress & density distribution during tableting	Mazel et al., 2016 [81]
		diametral compression of flat-faced tablets	Elastic stresses model	no data	4.2/0.25	Stress & strain distributions during diametral compression	
34	LAC, CS, MCC, L-HPC	diametral compression of flat-faced scored tablets	Elastic stresses model	no data	2.35/0.08	Stress & strain distributions during diametral compression	Okada et al., 2016 [88]
35	MCC	12 mm die, flat-faced punches	DPC	4-node (R)	*E* & *ν* expressed as function of relative density	Stress & density distribution during tableting	Baroutaji et al., 2017 [5]
36	ACP, SD-LAC	diametral compression of flat-faced tablets	Elastic stresses model	no data	4.4/0.25 for ACP & 3.7/0.23 for SD-LAC	Stress & strain distributions during diametral compression	Croquelois et al., 2017 [86]
37	ACP	12 mm die, concave-faced punches	DPC	no data	no data	Study of failure (lamination) mechanisms	Mazel et al., 2018 [100]
38	MCC	12 mm die, concave-faced punches	DPC	4-node (R)	no data	Study of failure (capping & chipping) mechanisms	Baroutaji et al., 2019 [96]
39	PCZ, MCC & ACP	9.525 mm die, concave-faced punches	DPC	4-node	no data	Stress & density distribution during tableting	Huang et al., 2019 [79]
40	MCC	11.28 mm die, flat- & concave-faced punches	DPC & Janssen–Walker model	no data	no data	Study of friction evolution during tableting	Mazel et al., 2019 [103]
41	MCC, LAC, CS	12 mm die, concave-faced punches	DPC	no data	1.803–3.321/0.1363–0.1774	Stress & density distribution during tableting	Takayama et al., 2019 [85]
42	LAC, ACP, MCC, CS	11.28 mm die, flat-faced punches	Linear viscoelastic	no data	no data	Study of the viscoelastic behavior during tableting	Desbois et al., 2020 [29]
43	MCC, LAC, ACT, MgSt	8.3 mm die, flat-faced punches	DPC & Perzyna model	no data	no data	Study of the viscoelastic behavior during tableting	Ohsaki et al., 2020 [110]
44	MCC, MgSt, LAC, NaCl	diametral compression of flat-faced tablets	Elastic stresses model	8-node	10.0/0.3	Stress & strain distributions during diametral compression	Radojevic et al., 2021 [91]
45	not applicable/theoretical study	diametral compression of various shape tablets	Elastic stresses model	20-node	no data	Stress & strain distributions during diametral compression	Yohannes and Abebe, 2021 [93]

^1^ Material abbreviations: LAC: Lactose monohydrate, MCC: Microcrystalline cellulose, CS: Corn starch, TEO: Theophylline, MgSt: Magnesium stearate, ACT: Acetaminophen, L-HPC: low-substituted hydroxy-propyl-cellulose, ASA: Acetyl salicylic acid, ACP: anhydrous calcium phosphate, SD: spray—dried, G: granulated, MAN: Mannitol, PCZ: Posaconazole, CPD: calcium phosphate dihydrate; ^2^ Model abbreviations: DPC: Drucker—Prager—Cap model; ^3^ Meshing: 4-node: four-node bilinear axisymmetric first order solid elements, 4-node (T): 4-node bilinear coupled displacement-temperature solid element, 8-node (T): 8-node trilinear coupled displacement-temperature solid element, 4-node (R): four-node axisymmetric first-order solid element with reduced integration, 3-node: 3-node linear axisymmetric triangular solid element, 20-node (R): 20-node “brick” element with reduced integration, 8-node: 8-node linear “brick” solid element, 10-node: 10-node tetrahedral solid element, 20-node: 20-node “brick” solid element; * *E* and *ν* values refer to the material at the end of compaction at its maximum relative density.

### 4.1. FEA in Compression

#### 4.1.1. Studies on the Stress/Density Distribution during Compression

Modeling stress/density distribution discloses important information for the prediction of failure mechanisms of the tablet during downstream processes (e.g., coating, packaging, transportation). Furthermore, the studies in Table 1 do not assign single values of Young’s modulus, E, and Poisson’s ratio, v, to the materials but rather report these parameters in relation to the relative density of the compressed powder. This is because the elastic behavior of a fully dense compacted material (i.e., at its highest relative density) emanates from interactions occurring at the atomic level [4]. Thus, E and v tend to increase rapidly as tablet evolves, and acquire maximum values at the end of compaction, where the condition of a fully dense compact is approached. The most indicative studies for the contribution of FEA in this application area are discussed.

Kadiri et al. (2005) [80] studied the axial stress distribution during MCC compaction using an analytical model derived by coupling Heckel with Janssen–Walker equations. They also studied the mechanical behavior of MCC tablets by numerical analysis (FEA) combined with the DPC model. Experimental data were produced by a press with an 11.28 mm die and flat-faced punches. Both the results of axial density obtained by the combined analytical model and by numerical analysis showed good qualitative agreement with the experimental. However, the experimental data were better predicted by the FEA model, which simulated both the compression and decompression stages, thus proving the superiority of FEA in pharmaceutical tableting.

Hayashi et al. (2013) [77] used the DPC model to study the residual stresses during compression of a powder mixture of LAC, MCC, and corn starch (CS), aiming for prediction of tablet tensile strength (*TS*) and disintegration time (*DT*). Experimental data were obtained by a press with an 8 mm die and flat-faced punches. The relationship between the residual stress distribution and tablet characteristics was investigated by multiple linear regression analysis (fitting index (*R*^2^) 0.992 for *TS* and 0.942 for *DT*, and root mean square error (RMSE) 0.080 for *TS* and 0.082 for *DT*. These results show that the residual stress distribution is a good estimator for *TS* and *DT*. Further prediction ability of tablet tensile strength and disintegration time based on the distribution of residual stresses was modeled by FEA and partial least squares regression analysis (PLS). RMSEc and RMSEp were 0.081 and 0.095, respectively, for the calibration model of *TS*, and 0.085 and 0.1 for *DT*. It was concluded that both *TS* and *DT* could be accurately predicted by the residual stress distribution modeled by FEA, suggesting that the latter can be used as CQA (critical quality attribute) in pharmaceutical development.

#### 4.1.2. Effect of Punch Shape on the Stress/Density Distribution during Tableting

The majority of commercially available pharmaceutical tablets have convex surfaces to ease swallowing. This shape is likely to affect the stress distribution within the tablets and the powder during compression compared to flat faced tablets [90,101]. From Table 1, it appears that the number of studies investigating the effect of the punch shape on the stress distribution is increasing since 2018, proving that FEA is still gaining interest and has not revealed its full potential in pharmaceutical tableting. Some indicative studies are discussed below.

Mazel et al. (2015) [28] studied the effect of concave-shaped punches on the residual die-wall stresses during decompression of tablets of anhydrous calcium phosphate (ACP) based on the DPC model. Experimental data were obtained using a press with an 11.28 mm die and flat or concave-faced punches producing convex tablets. The results showed, during compression, a lower maximal die-wall pressure and a higher residual die-wall pressure for the convex tablets compared to the flat tablets. Moreover, for the biconvex tablets, a temporary increase in die-wall pressure at the end of decompression was recorded. FEA indicated that this increase was due to a gradual loss of contact between the concave punch face and the tablet from the side to the center. Thus, a temporary increase in the die-wall pressure and the development of shear stress between the convex and the flat part of the tablet may arise, which explains the capping tendency of convex tablets.

Otoguro et al. (2015) [82] studied the distribution of residual stress in tablets of a mixture of acetaminophen (ACT), LAC, CS, MCC, low-substituted hydroxypropyl cellulose (L-HPC), and magnesium stearate (MgSt) powders based on the DPC model. Experimental data were acquired by a press fitted with 12 mm die (long dimension) and flat- or concave-faced punches. The stress distribution was studied after application of two different compression forces: 4 and 8 kN, at two MgSt levels, 0.5 and 2.0%. For flat tablets, weak positive residual shear stresses, $\tau_{xy}$, were recorded on the die walls, decreasing from the top and the bottom of the tablet towards its center. For the convex tablets, strong positive residual $\tau_{xy}$ stresses were recorded on the upper side and the intermediate part between the die wall and the center of the tablet. For both tablet shapes, negative x-axial $\sigma_{xx}$ stress values were observed, implying that $\sigma_{xx}$ stresses always act from the die wall toward to the center of the tablet. Weak residual stress in the y-axial direction, $\sigma_{yy}$, of the flat tablet were also recorded, whereas an upward force remained at the center of the convex tablet. MLR (multiple linear regression) analysis was also employed and gave accurate prediction of the mechanical properties of the tablets. However, MLR failed to predict the dissolution performance of ACT, implying that dissolution is complex and does not depend only on the stress distribution within the tablet.

Baroutaji et al. (2019) [96] studied the effects of geometrical parameters on the compression of convex-faced tablets by applying experimental design (DoE) and response surface methodology (RSM) to compression responses computed by FEA in order to optimize tableting. Relationships were established between the diameter and radius of curvature with the friction coefficient, residual die pressure, relative density variation, and relative shear stress. The shape of the convex tablets was optimized. The FEA model was based on the DPC model. Experimental data were obtained by a press fitted with a 12 mm die and flat- or concave-faced punches. It was found that both the geometric parameters and friction coefficient significantly affected the compression responses. After optimizing these parameters for convex tablet (CT), the compression responses were compared with those of a flat tablet (FT). As Figure 5 shows, flat-faced tablets (FT) exhibited lower density variation than the CT. The relative shear stress of the FT was 47% smaller than that obtained for the optimal CT tablet. Additionally, from Figure 5c, it is seen that the evolution of the die-wall pressure is a function of the radial pressure for FT and CT tablets. Furthermore, larger residual die press evolved during the production of FT tablets than during that of CT tablets, and it was concluded that, if there is no occurrence of capping or chipping, a better performance is expected from CT than FT tablets during downprocessing (coating, packaging, transportation).

Takayama et al. (2019) [85] studied the effect of concave-shaped punches on the residual stresses recorded during compression of a mixture of LAC, MCC, and CS powders based on the DPC model aiming for prediction of tablet *TS* and *DT*. Experimental data were obtained by a press fitted with a 12 mm die with flat- or concave-faced punches. A clear difference in the residual stress distributions between the flat and convex tablets was reported. High residual stresses were observed in the convex tablets, but low in the flat tablets. Moreover, elastic-net (ENET) regression was employed to sparse the model and identify specific stress sites in the tablets affecting the *TS* and *DT*. Both the quantity and the direction of residual stresses acting at specific sites close to the die wall were crucial in the convex tablets for prediction of *TS* and *DT*. Such tendency was not observed for flat tablets. However, the linkage of residual stresses at specific sites still needs clarification, and further studies are required to understand the mechanism of how these forces affect the *TS* and *DT*. In conclusion, FEA coupled with ENET regression was able to study in depth the relationships between the residual stress distribution and the characteristics of convex tablets.

#### 4.1.3. Studies on Failure Mechanisms

Despite the long history of pharmaceutical tableting, the process still presents major challenges since, at the extremely high-speed operating, modern machines’ defects, such as capping, lamination, and chipping, are not uncommon. From Table 1, it appears that, only in a few studies, the failure mechanism is investigated utilizing FEA. Indicative studies are discussed below.

Han et al. (2008) [75] studied the von Mises stress distribution during compression of MCC employing a modified DPC model to describe the non-linear elastic behavior of the powder during decompression. Experimental data were acquired by a press fitted with an 8 mm die/flat-faced punches. For the assessment of die-wall friction during tablet ejection by the FEA model, two friction coefficient values (μ) were used. Zero (μ) was set to model the case of lubricated punch and die and (μ) = 0.2 was set for un-lubricated tools. Non-uniform von Mises stress distribution was reported in the un-lubricated case. During compression/decompression, high von Mises stress regions were detected at the top corner, while low von Mises stress regions were detected at the bottom corner. After ejection, high stress regions were formed close to the die edge, which were attributed to the radial elastic recovery of the tablet. Han et al. (2008) [75] linked these non-uniform stress distributions to chipping, capping, and lamination in the tablets. X-ray tomography was further employed to explore this link. As can be seen in Figure 8 of their publication, there is agreement between the numerical von Mises stress patterns at the ejection stage and the cracking patterns observed by X-ray tomography. As the tablet exits the die, cracks initiate at the top corner, where highest von Mises stresses are predicted by FEA. Therefore, it is concluded that modeling compression, decompression, and ejection stages by FEA provides a quantitative assessment of tablet quality and prediction of failure probability.

In another work of Mazel et al. (2018) [100], the lamination of biconvex tablets of anhydrous calcium phosphate (ACP) was studied with the aid of FEA based on a modified DPC model. Experimental data were acquired using a 11 mm die (longest dimension) with concave punches. Numerical simulation of the data with FEA pointed to lamination of biconvex tablets due to tensile stresses developing at the center of the tablet, which are induced by the residual die wall pressure in the biconvex tablet shape. As the crack is formed at the center of the tablet, it may not propagate immediately. Thus, failure may remain undetected by external visual examination. However, X-ray tomography was able to detect central cracks inside the tablet even without breakage. Therefore, FEA can act as a pre-diagnostic tool to avoid undetected ongoing cracks, which could have dramatic consequences during further tablet processing.

#### 4.1.4. Studies on Die-Wall and Powder Friction

Tableting is sometimes referred to as uniaxial die compression because pressure is applied to the powder by the vertical movement of one or two punches. Die-wall pressure and friction acting during the ejection stage play an important role for the final tablet quality. Experimental studies indicate that friction depends on several factors, including: contact pressure, local powder density, sliding velocity, sliding distance, temperature, and wall roughness [10]. There are a number of studies on the friction between the powder and tooling studied with the aid of FEA (Table 1). Two indicative studies are discussed below.

Michrafy et al. (2004) [104] studied the axial density profile of flat-faced MCC tablets compressed in un-lubricated die by applying FEA together with the DPC and Janssen–Walker models. Die wall friction coefficient and other material model parameters were estimated from the experimental data. The axial density distribution was computed from simulation of the compression/decompression results. Comparison with previously published data indicated that it is possible to correlate powder die-wall friction and a simple loading path with the density distribution in the compact, proving the versatility of FEA.

Mazel et al. (2019) [103] studied the influence of friction between the compressed MCC powder and the tooling on the evolution of die-wall pressure for the cases of flat-faced and concave punch shapes. Experimental data were acquired by a press fitted with a 11.28 mm die (long dimension). Numerical modeling via FEA with implementation of DPC and Janssen–Walker models was used to confirm and interpret the experimental trends. The results showed that, for flat punches, the stress evolution is mainly driven by the die wall/powder friction. However, for concave punches, the punch/powder friction had a significant effect on the evolution of die-wall pressure. Consequently, sticking during compression due to the high powder-punch friction coefficient may lead to increased die-wall pressure. Therefore, variation in the lubrication conditions during compression could have an effect on tablet final properties. The results of the present study were in contradiction with results that can be found in the literature reporting no influence of the external lubrication on the die-wall pressure [111].

Moreover, Sinka et al. (2004) [105] analyzed tablet compaction by FEA of the density distribution in convex tablets and demonstrated that the same compaction force under low friction produced higher average tablet relative density of MCC tablets, which, in turn, affected the failure mode and strength.

#### 4.1.5. Studies on Temperature Evolution during Tableting

The increase in temperature during pharmaceutical tableting is widely recognized and has been of concern since 1968 [112] on the characteristics of the tablet. Among others, it may affect the compressibility and tablet strength, the performance of lubricants, the die wall friction, and, hence, ejection force, as well as may induce physicochemical changes (e.g., stability, polymorphism, crystalline state) [108]. From Table 1, it appears that only three articles report FEA for the study of temperature evolution during tableting. The work by Klinzing et al. (2010) [106] is discussed below.

Temperature evolution during tableting of MCC was studied with the aid of FEA and DPC model coupled with thermomechanical analysis. Experimental data were acquired by using a press fitted with a 9.525 mm die and concave punches. Figure 6 shows experimental density values taken by X-ray microtomography (mCT) against FEA predictions. Both experimental and simulation analysis results clearly show a gradient of higher density in the areas where the tablet is in contact with the die and lower density in the middle of the tablet. The prediction of porosity and temperature distribution by the FEA model were found to be in agreement with the mCT results and infrared camera measurements of surface temperature of the ejected tablet. Moreover, the fact that FEA model was calibrated with data from differently shaped tablets (cylindrical, flat-faced) enhances the predictive capability of FEM.

#### 4.1.6. Studies of the Viscoelastic Behavior during Tableting

Compaction speed can vary between commercially used and developmental tableting machines, which, for the industrial rotary machines, can be extremely high. It is well known that any dependency of the compression of materials on compression speed expressed as strain rate sensitivity (SRS) can have consequences on the final tablet quality attributes. SRS describes the viscoelastic or viscoplastic behavior of pharmaceutical powders that exhibit time-dependent elastic or plastic deformation, respectively. Three studies have applied FEA for the study of the viscoelastic behavior of pharmaceutical powders during tableting. The work of Ohsaki et al. (2020) [110] is discussed below.

A powder mixture consisting of MCC, lactose (LAC), acetaminophen (ACT), and magnesium stearate (MgSt) was compressed at different speeds and the process was studied with the aid of FEA based on the DPC and Perzyna models [113]. Experimental data were acquired by a press fitted with 8.3 mm die/flat-faced punches. The DPC–Perzyna model parameters were determined experimentally from compaction tests, unconfined compression tests, and tensile tests. The calculated loading curves agreed with the experimental data obtained under different compression speeds. High speeds resulted in less plastic deformation but more residual stress. It was demonstrated that FEA adopted with the DPC and Perzyna models was useful for the analysis of tableting at variable speeds.

### 4.2. Application of FEA in Diametral Loading Test of Mechanical Strength

#### Stress & Strain Distributions during Diametral Loading

The mechanical strength of pharmaceutical tablets is an important part of research programs, aiming to understand the mechanism by which tablet ingredients stick together to form a strong tablet, and also to reveal the important characteristics of the ingredients relevant to bonding. The diametral compression test is the most frequently used test for the evaluation of the mechanical strength of round pharmaceutical tablets. A full stress analysis is performed, which allows calculation of the tensile stress at failure from the magnitude of the breaking load. However, for non-simple shapes (e.g., capsule shape, lozenge, convex shapes) a simple analytical solution is not available and stress analysis by simulation is required. For a complex stress analysis of different tablet shapes and elucidation of stress and strain distribution, FEA has been extensively implemented. Stanley (2001) [114] emphasized the need to deeply understand the stress distribution induced in a loaded specimen in order to convert the breaking load into a strength value. Although FEA has also been employed for the study of stress distribution induced by other types of mechanical tests, e.g., three point bending [94,95] and axial crushing test [99], the diametral loading compression has become the established method in pharmaceutical technology, and, for this reason, from the studies listed in Table 1, only those related to this test are discussed.

Pitt and Heasley (2013) [65] sought analytical solutions and mathematical equations for the calculation of the tensile strength of elongated tablets with the aid of FEA. Indeed, 10-mm convex- and capsule-shaped tablets were used for a hypothetical material with 0.02 GPa Young’s modulus. In Figure 7, the 3D model of a circular tablet under compression is presented, showing positive and negative tensile stress in the *x* direction. The results of stress analysis showed that, as the tablet was getting more elongated from a standard circular tablet shape, the tensile stress approached a limiting value. This value was achieved once the length to width ratio exceeded 1.7/1, which encompasses most of today’s pharmaceutical tablets. In addition, this value approximated 2/3 of the corresponding value of a circular tablet. Thus, a modification to the equation by Pitt et al. (1988) [70] was proposed in order to adapt this case to a convex-faced elongated tablet. Application of the modified equation to commercial tablets of various shapes was also demonstrated.

In the same year, Podczeck et al. [89] investigated analytical mathematical solutions for the calculation of the tensile strength of anhydrous dextrose convex-shaped tablets. FEA was employed to model and evaluate the tensile stress at failure of convex tables with central cylindrical part to total tablet thickness ratio (W/D), between 0.06 and 0.50, and face-curvature ratio (D/R), between 0.25 and 1.85. Both elastic and elasto-plastic deformation behavior was considered. The results of 80 individual simulations showed that the tensile failure stress of convex tablets can be calculated from the standard Brazilian test equation for flat tablets, and this was valid for all combinatory (W/D) ratios between 0.06 and 0.50 and (D/R) ratios between 0.00 and 1.85. For the combination (W/D) 0.50 and (D/R) 1.85 or 1.43, and for the combination of (W/D) 0.40 or 0.30 with (D/R) 1.85, the modified equation of Pitt and Heasley [65] was a better alternative.

Podczeck et al. (2014) [90] investigated the influence of the position of a breaking (“score”) line on the tensile failure and stress/strain distributions for flat, round, and bevel-edged tablets subjected to diametral compression test using FEA model. Various breaking line test positions at angles of 0°, 22.5°, 45°, 67.5°, and 90° relative to the loading plane were studied. The theoretical investigation referred to a material of E = 2.58 GPa and a v = 0.35. Both elastic and elastoplastic deformation behavior were taken into account. From Figure 8, it appears that the results obtained for fully elastic and elastoplastic tablets were fairly similar. However, large differences were observed in stress distributions depending on the position of the breaking line. The stress at failure was predicted to be similar for tablets tested at an angle of 45° or above, whereas, at lower test angles, the predicted breaking loads were up to three times larger. According to stress distributions, not all breaking line angles would result in clean tensile failure. A comparison of the theoretical results with experimental data, however, did not confirm the differences in the predicted breaking loads. On the other hand, the experimental results confirmed differences in the way that tablets broke depending on the position of the breaking line. The results of the study suggest that breaking loads applied to scored tablets cannot be converted into tablet tensile strength values. Furthermore, comparisons between different tablets or batches should carefully consider the orientation of the breaking line with respect to the loading plane as the failure mechanisms appear to vary.

Croquelois et al. (2017) [86] reevaluated the stress concentration factor (SCF) in the case of cylindrical and flattened tablets [81] with holes, during the diametral compression test, with the aid of FEA; 11-mm tablets were tested for two different materials: acetaminophen (E = 4.4 GPa, v = 0.25) and spray dried lactose (E = 3.7 GPa, v = 0.23). It was reported that the value of SCF was nearly independent of the hole size when the ratio of the hole and the tablet diameters was less than 0.1. Nevertheless, the experimental results presented in this work showed that the failure load on a compact varied with the hole size. These results were attributed to changes in the stress distribution around the hole as the hole size changed. Criteria such as the average stress criterion, which takes into account the stress distribution, made it possible to explain the influence of the hole size on the breaking load.

## 5. Outlooks

There is a great need for a deeper understanding of the tableting process, which can be gained only by application of new research tools. As shown in the present review paper, finite element analysis (FEA) combined with constitutive models has the capability to fulfil this expectation by providing new information and knowledge on the mechanisms involved leading to the production of tablets with improved performance. Nevertheless, the increasing complexity and divergence of FEA software generate ambiguities regarding its application. Therefore, there is a need to standardize the implementation of FEA in the field of pharmaceutics.

In the engineering field, two main regulatory bodies, the American Society for Testing and Material (ASTM) and the International Association for the Engineering Modelling, Analysis and Simulation (NAFEMS) provide some guidelines (ASTM F2996, ASTM F2514, and ASTM F3161) for the implementation of FEA in pharmacy and medicine. However, these do not address the pharmaceutical tableting process. As shown in Table 1, in some cases (indicated as ‘no data’), components of the FEA model, such as meshing, Young’s modulus, and Poisson ratio, are not mentioned, and, when reading the publications, boundary conditions and material properties are not given in detail, making it hard to interpret the results. Therefore, more collaborative efforts have to be made by scientists working in this field in order to improve the communication of the data and scientific outcomes that would open up and encourage further research in this area.

## 6. Conclusions

Finite element analysis (FEA) coupled with constitutive models has been successfully implemented to predict powder behavior during tablet compression with differently shaped pharmaceutical tablets. FEA, together with the Drucker–Prager CAP (DPC) model, is more often employed and has been shown to be a practical, potent, resource-saving, and reliable research method that can generate crucial information about complex physical phenomena occurring during tableting, such as: stress and density distribution during compression and during testing of the compact by diametral loading, tablet failure mechanisms and tablet mechanical strength, temperature evolution during compression, and frictional forces between the compressed material and compression tools. The value of FEA in the pharmaceutical tableting field has been recognized in the last two decades, and FEA is expected to attract more interest from researchers as its potential and feasibility are realized. It is a simple tool and, only by means of a computer and the appropriate software, it can simulate and predict important aspects of the compression of pharmaceutical powders, crucial to their quality and performance.

## Figures and Tables

**Figure 1 pharmaceutics-14-00673-f001:**
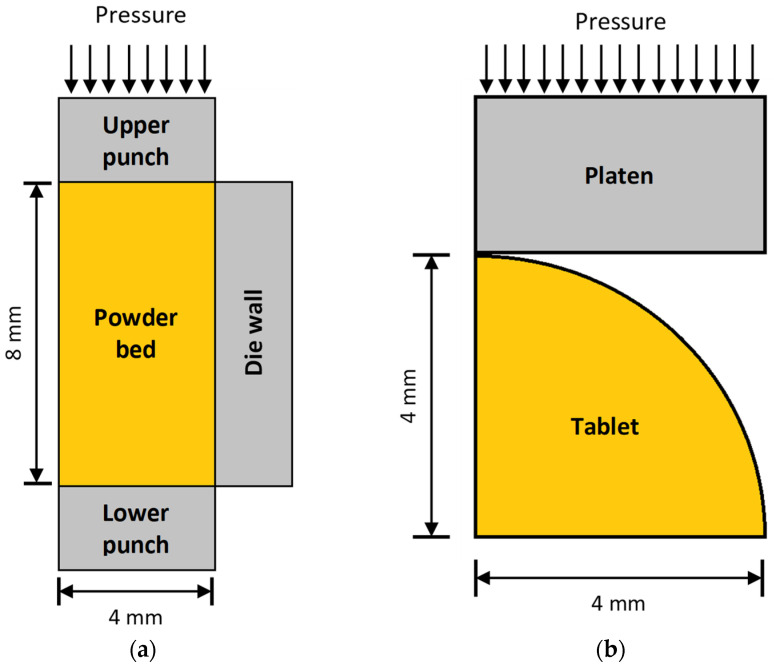
Symmetric geometric 2D model (**a**) for powder compression and (**b**) for diametral loading test of tablet mechanical strength.

**Figure 2 pharmaceutics-14-00673-f002:**
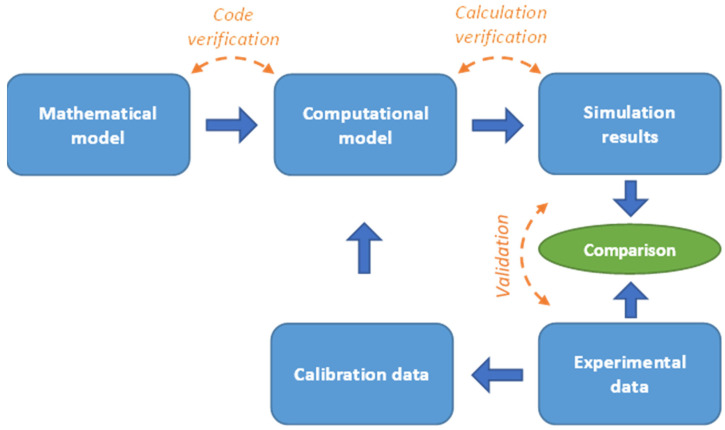
Flowchart for the application, verification, and validation of a finite element analysis (FEA) model.

**Figure 3 pharmaceutics-14-00673-f003:**
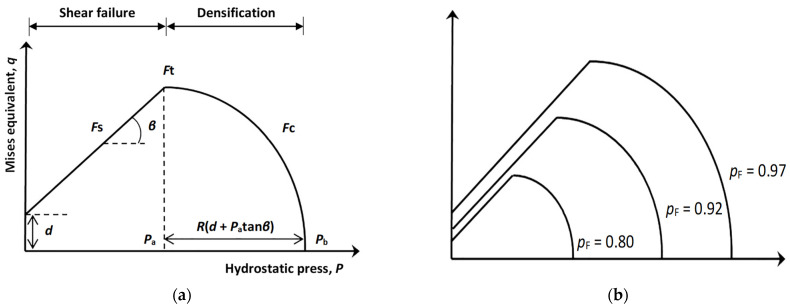
(**a**) The Drucker–Prager Cap (DPC) model and its parameters and (**b**) family of DPC models for different levels of relative density over a range of compaction.

**Figure 4 pharmaceutics-14-00673-f004:**
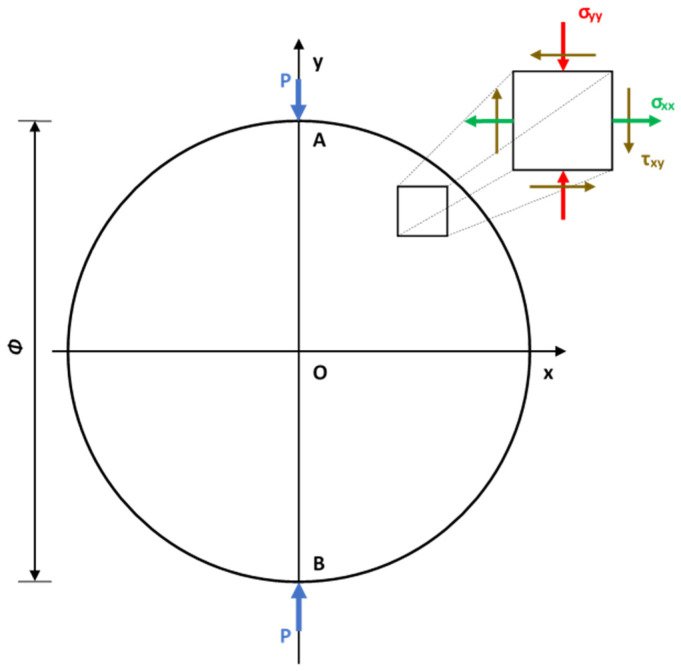
Elastic stress field developed in a flat-faced tablet during diametrical compression.

**Figure 5 pharmaceutics-14-00673-f005:**
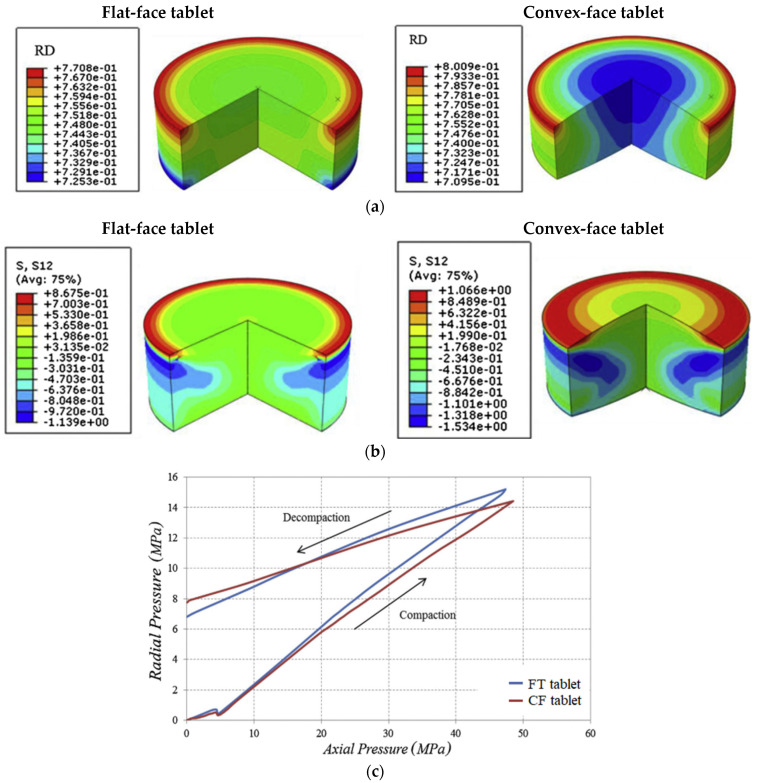
Comparison of flat- and optimal convex-faced tablets: (**a**) relative density (RD) distribution, (**b**) shear stress distribution within the tablets as they eject from the die, and (**c**) plot of radial pressure vs. compaction pressure (Adapted with permission from Ref. [96]).

**Figure 6 pharmaceutics-14-00673-f006:**
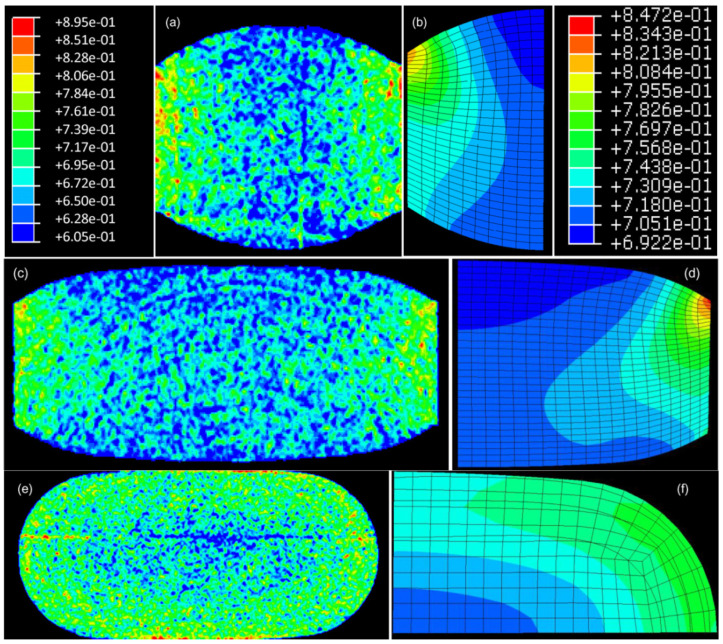
Microcomputed tomopraphy relative density cross-sections (left column) and correlating simulation cross-sections (right column) of a convex tablet (color contour scales represent relative density values). Reprinted with permission from Ref. [106].

**Figure 7 pharmaceutics-14-00673-f007:**
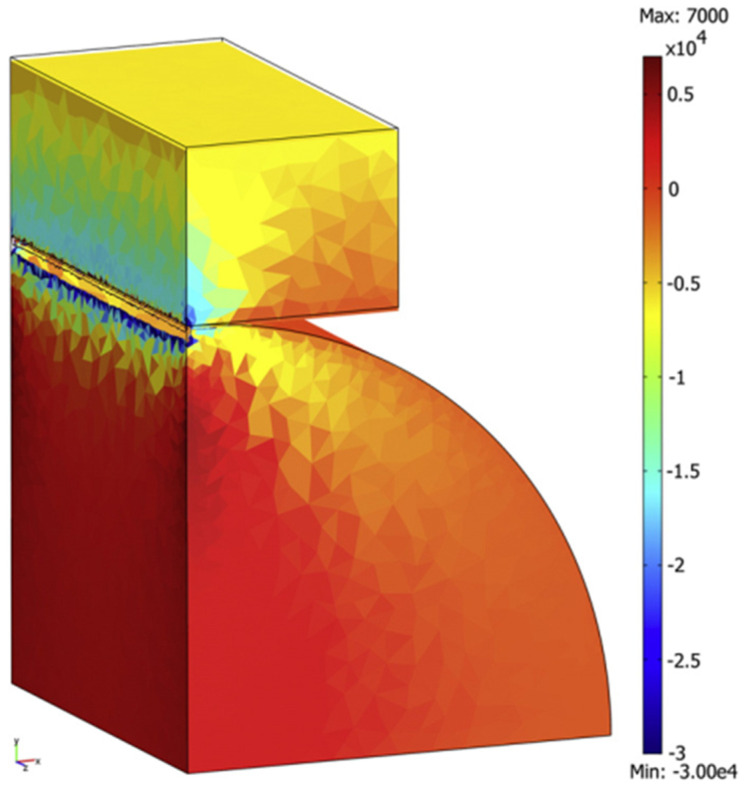
3D model of circular tablet under compression showing positive and negative tensile stress in the x direction. Reprinted with permission from Ref. [65].

**Figure 8 pharmaceutics-14-00673-f008:**
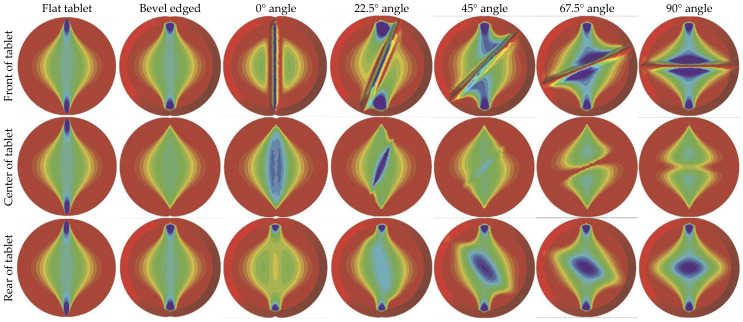
x-axial stress distribution in elastic tablets with breaking (“score”) line (Adapted with permission from Ref. [90]).

## Data Availability

Not applicable.

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
