# Peer review of "Finite Element Analysis and Modeling in Pharmaceutical Tableting"

_pharmaceutics, 2022, doi:10.3390/pharmaceutics14030673_

Round 1
Reviewer 1 Report
The manuscript “Finite Element Analysis and Modeling in Pharmaceutical Tableting” is a good example of a comprehensive review in the specific scientific area, which will be definitely fruitful for the potential reader interested in a prediction of the tablets’ formation regularities and properties of the resulting product that can help to avoid a number of preliminary experiments and tests. The paper provides rather deep knowledges on general principles of tablets’ modelling, existing challenges, the role of the finite element analysis (FEA) in the field and examples of its application. The review is logically built and created using reputational databases. It covers both historical and state-of-the-art aspects of the FEA implementation in the modelling pharmaceutical tableting. There is no self-citation at all but the way of the review preparation shows that the authors are experts in the FEA and its utilization in pharmaceutics. The manuscript is well written, understandable for the audience and does not need English style or grammar corrections.
There are just a few recommendations for the authors before accepting the manuscript for publication.
- This would be very nice to see images of 3D geometries (not only 2D) and meshes of tablets that are typically used for modelling in Section 2. The authors demonstrated some examples in the 4th Section but visualization of some standards will be a plus.
- It is suggested to mention basic software utilized for the tablets’ modelling to make the review more informative for the beginners.
- The last section encloses the review in a short enough way while this is expected to read the authors’ outlook regarding future development and perspectives of the FEA method in pharmaceutics.
Author Response
MDPI Pharmaceutics
Manuscript ID: pharmaceutics-1641592
Reviewer #1:
We are very thankful to the Reviewer for the constructive comments and suggestions. The questions raised have been responded point-by-point. The added text in the revised manuscript is marked in red.
Q1: “The manuscript “Finite Element Analysis and Modeling in Pharmaceutical Tableting” is a good example of a comprehensive review in the specific scientific area, which will be definitely fruitful for the potential reader interested in a prediction of the tablets’ formation regularities and properties of the resulting product that can help to avoid a number of preliminary experiments and tests. The paper provides rather deep knowledges on general principles of tablets’ modelling, existing challenges, the role of the finite element analysis (FEA) in the field and examples of its application. The review is logically built and created using reputational databases. It covers both historical and state-of-the-art aspects of the FEA implementation in the modelling pharmaceutical tableting. There is no self-citation at all but the way of the review preparation shows that the authors are experts in the FEA and its utilization in pharmaceutics. The manuscript is well written, understandable for the audience and does not need English style or grammar corrections.
There are just a few recommendations for the authors before accepting the manuscript for publication..”
Response: Thank you very much for your comments.
Q2: “This would be very nice to see images of 3D geometries (not only 2D) and meshes of tablets that are typically used for modelling in Section 2. The authors demonstrated some examples in the 4th Section but visualization of some standards will be a plus.”
Response:
Image of a 3D model of circular tablet under compression has been added in section 4.2 for better visualization as suggested by the reviewer (lines 784-785 and 794-795 in the revised).
Q3: “It is suggested to mention basic software utilized for the tablets’ modelling to make the review more informative for the beginners.”
Response: Basic software was already included in the original submission, please refer to lines 210-214.
Q4: “The last section encloses the review in a short enough way while this is expected to read the authors’ outlook regarding future development and perspectives of the FEA method in pharmaceutics.”
Response: A section of 21 lines of outlook regarding future development and perspectives of FEA has been added in the revised (lines 844 to 864) a suggested. The need for regulatory bodies to standardize the implementation of FEA in pharmaceutics is emphasized.
Reviewer 2 Report
It's an excellent revision work, I don't know if maybe it's a bit long. Detailed understanding of the compression process is an important prerequisite for making quality tablets, so it is imperative to develop accurate and cost-effective predictive models for the different unit operations and process flow diagrams. These models are already used to predict product quality and maintain desired manufacturing efficiency. Thus, the residence time distribution has been a widely used tool to characterize the degree of mixing in pharmaceutical operations. It is true that it has been widely described in the literature, but the implementation, execution, and evaluation of these studies have not been standardized by regulatory agencies and, therefore, can generate ambiguity regarding their application. The authors could indicate what could be done about this issue.
The authors show that the Prager CAP (DPC) model is practical, powerful and resource-saving, which means that crucial information can be generated regarding the physical phenomena that occur during tablet production. It is a very interesting work with great potential applicability in the field of pharmaceutical technology, since it is a simple tool to predict important aspects of the compression of pharmaceutical powders, crucial for the final quality of the drug.
There would only be two questions, because it is a very well written and discussed work.
1.- The extension can be a bit long, it would be preferable to summarize the number of pages.
2.- Detailed understanding of the compression process is an important prerequisite for manufacturing quality tablets, but the implementation of these studies have not been standardized by regulatory agencies and therefore may lead to ambiguity as to their application, the authors could indicate what could be done about it.
Author Response
MDPI Pharmaceutics
Manuscript ID: pharmaceutics-1641592
Reviewer #2:
We are very thankful to the Reviewer for constructive comments and suggestions. The questions raised have been responded by a point-by-point manner, and added text is marked in red in the revised manuscript.
Q1: “It's an excellent revision work, I don't know if maybe it's a bit long. Detailed understanding of the compression process is an important prerequisite for making quality tablets, so it is imperative to develop accurate and cost-effective predictive models for the different unit operations and process flow diagrams. These models are already used to predict product quality and maintain desired manufacturing efficiency. Thus, the residence time distribution has been a widely used tool to characterize the degree of mixing in pharmaceutical operations. It is true that it has been widely described in the literature, but the implementation, execution, and evaluation of these studies have not been standardized by regulatory agencies and, therefore, can generate ambiguity regarding their application. The authors could indicate what could be done about this issue.
The authors show that the Prager CAP (DPC) model is practical, powerful and resource-saving, which means that crucial information can be generated regarding the physical phenomena that occur during tablet production. It is a very interesting work with great potential applicability in the field of pharmaceutical technology, since it is a simple tool to predict important aspects of the compression of pharmaceutical powders, crucial for the final quality of the drug.”
Response: Thank you very much for your comments.
Q2: “There would only be two questions, because it is a very well written and discussed work.
1.- The extension can be a bit long, it would be preferable to summarize the number of pages.
2.- Detailed understanding of the compression process is an important prerequisite for manufacturing quality tablets, but the implementation of these studies have not been standardized by regulatory agencies and therefore may lead to ambiguity as to their application, the authors could indicate what could be done about it.”
Response:
- The structure of the paper was organized so as to balance the feedback on the theory, the steps of FEA and the associated constitutive models with information on the applicability to tablet compression and quality prediction. If the reviewer considers appropriate to reduce the size of some section it woud help us to know. Otherwise we prefer to keep the content of the paper in its present form.
- A section of 21 lines regarding future development and perspectives of FEA has been added in the revised (lines 844 to 864). The need for regulatory agencies to standardize the implementation of FEA in pharmaceutics is emphasized.
Reviewer 3 Report
Dear Authors,
The submitted manuscript is a very interesting example of a review article taking into consideration finite element analysis (FEA) as a computational method providing numerical solutions and mathematical modeling of complex physical phenomena that evolve during compression tableting of pharmaceutical powders.
Introduction part including description of finite element analysis is well prepared.
Finite Element Modeling and Analysis are well and properly described. The Reviewer was really impressed by the detailed description of the performed review studies.
Is that possible to generalize the collected data using statistical analysis to mathematically prove that Finite Element Modeling represent an important part in the pharmaceutical technology and design?
Author Response
MDPI Pharmaceutics
Manuscript ID: pharmaceutics-1641592
Reviewer #3:
We are very thankful to the Reviewer for constructive comments and suggestions. The questions raised have been responded by a point-by-point manner, and added text is marked in red in the revised manuscript.
Q1: “Dear Authors,
The submitted manuscript is a very interesting example of a review article taking into consideration finite element analysis (FEA) as a computational method providing numerical solutions and mathematical modeling of complex physical phenomena that evolve during compression tableting of pharmaceutical powders.
Introduction part including description of finite element analysis is well prepared.
Finite Element Modeling and Analysis are well and properly described. The Reviewer was really impressed by the detailed description of the performed review studies.”
Response: Thank you very much for the comments.
Q2: “Is that possible to generalize the collected data using statistical analysis to mathematically prove that Finite Element Modeling represent an important part in the pharmaceutical technology and design?”
Response: We have thought of performing statistical analysis to prove mathematically the value of FEA in pharmaceutical technology. Since the currently submitted review paper is already too long, we think that further addition would hamper readability. For this reason we plan to prepare a separate piece of work on the statistical evalaution of the data presented in the paper.